# Endemic and zoonotic cycles of cutaneous leishmaniasis depend on vector feeding preferences: An epidemiological model for Southeastern Mexico

Gerardo Martín[1]*, Edgar J. González[2], Elsy Nallelli Loría-Cervera[3], Ana Celia Montes de Oca-Aguilar[3]*

**1** Departamento de Sistemas y Procesos Naturales, Escuela Nacional de Estudios Superiores unidad Mérida, Universidad Nacional Autónoma de México, Mérida, México, **2** Departamento de Ecología y Recursos Naturales, Facultad de Ciencias, Universidad Nacional Autónoma de México, Mexico City, México, **3** Laboratorio de Inmunología, Centro de Investigaciones Regionales "Dr. Hideyo Noguchi", Universidad Autónoma de Yucatán, Mérida, México

\* gerardo.mmc@enesmerida.unam.mx (GM); ana.montesdocaa@gmail.com (ACMdO-A)

## Abstract

### Introduction

Cutaneous leishmaniasis (CL) is a neglected tropical disease caused by *Leishmania spp.* parasites. It poses significant public health challenges among economically marginalized communities in endemic regions like Southeastern Mexico. Here, we developed a mathematical model to describe the transmission dynamics of CL in the Yucatan Peninsula region, focusing on two key sand fly vectors: *Lutzomyia cruciata* and *Bichromomyia olmeca*.

### Methods

Transmission was modelled as frequency dependent with Susceptible-Infected dynamics for vectors and rodents, and Susceptible-Exposed-Infected-Susceptible for humans, and accounts for the impact of available blood meals on vector populations. The model was parameterized from published literature for vector and reservoir growth rates and carrying capacities, transmission efficiency, and vector feeding preferences.

### Result

The simulations highlight the importance of both vector species in CL transmission, with *Lu. cruciata* showing a higher preference for human blood, while *Bi. olmeca* is more frequently associated with rodents. Sensitivity and scenario analyses reveal that the system is highly sensitive to transmission rates and vector feeding

**Data availability statement:** All data and model code are available in the GitHub repository https://github.com/gerardommc/Leish-transmission.git, and its Zenodo archive https://doi.org/10.5281/zenodo.17717116.

**Funding:** Funding was provided by the Secretaría de Ciencia, Humanidades, Tecnología e Innovación, project number CBF-2025-I-2041 (granted to GM, EJG, ENLC and ACMOA). The funders had no role in study design, data collection and analysis, decision to publish, or preparation of the manuscript.

**Competing interests:** The authors have declared that no competing interests exist.

preferences, suggesting that *Bi. olmeca* is necessary to maintain infection among reservoirs, while *Lu. cruciata* contributes to zoonotic transmission.

## Conclusions

Overall, we emphasize that vector species have different roles on leishmaniasis epidemiology due to feeding preferences. Therefore, roviding funding to improve basic knowledge into the epidemiology of this disease, will improve our understanding of its dynamics. Once achieved, policymakers can develop targeted interventions to reduce CL incidence in affected regions to improve the outcomes of public health interventions.

### Author summary

Cutaneous leishmaniasis (CL) is a skin disease caused by parasites and spread by the bites of infected sand flies. It affects vulnerable populations in rural and marginalized communities, such as those in Southeastern Mexico. Although it can cause long-lasting skin sores and social stigma, many aspects of how this disease spreads in the region remain poorly understood. We developed a mathematical model to better understand the transmission of CL in the Yucatan Peninsula, focusing on two sand fly species that feed, at different degrees, on both humans and wild animals. Our results show that the two sand fly species play different roles in disease transmission: one species maintains wild animal infection, while the other transmits the disease to humans. These results derive from data scattered across the study region, which carries significant uncertainty, and represent important information gaps. These limitations highlight the need for further research on how local sand fly populations interact with human and animal hosts, thus improving modelling efforts for disease control.

## Introduction

Cutaneous leishmaniasis (CL) is a neglected tropical disease caused by *Leishmania spp*. protozoan parasites [1]. Clinically, CL is characterized by chronic skin lesions which may develop into face disfigurement [2]. In endemic areas, CL is a serious public health problem contributing with 0.58 disability adjusted life years (DALYs) per 100,000 people [3], especially among economically marginalized communities [4]. Although CL typically has very low mortality, its DALYs have a significant contribution to the poverty trap [4]. The annual CL incidence in the Americas is estimated to fluctuate between 187,200 and 307,800 cases [5], with Brazil, Colombia and Venezuela having the highest burden [6]. Controlling leishmaniasis transmission has been limited because it is a zoonotic disease involving multiple wild non-human mammalian reservoirs and insect vector species of the Phlebotominae subfamily (Diptera:

Psychodidae) [7]. Thus, multiple complications arise from our poor understanding of its transmission cycles including the interactions between various vectors and reservoir species.

In the Neotropics, humans typically acquire CL near tropical forests, notably while working on agriculture, timber and non-timber forest products or during activities related to land conversion. Thus, working, living or settling in forest stands is the main risk factor for acquiring CL in the Americas [8]. In this way, humans become accidental, frequently dead-end hosts [9], and therefore they are considered irrelevant for the long-term persistence of CL in endemic areas [10].

The Americas are experiencing regional changes in the epidemiology of CL and mucocutaneus leishmaniasis (ML). In the past years (2016–2022), there has been a decline in the number of cases and incidence of CL and ML in 12 of the 17 countries considered to have high endemic transmission [6,11–17]. In 2016 alone, 48,915 cases were reported, of which 94–97% were CL, with an incidence of 21.71/100,000 habitants, while 41,617 cases and an incidence of 18.78/100,000 habitants were recorded in 2019 [6]. However, Mexico, together with four other countries that do not belong to the high transmission ranking, exhibit a non-linear increase from 76 to 146% in CL-ML cases [6,11–13,17–19]. In Mexico, CL accounts for 99% of the cases, of which ~60% are located in the biogeographic province of the Yucatan Peninsula [20].

The transmission of *Leishmania mexicana mexicana* in the Yucatán Peninsula is hyperendemic [21]. Currently several outbreaks of CL have been identified in areas with no historical records [20,22,23]. In this region, rural Mayan communities are the most affected since they have established a strong historical relationship with the tropical forest [24,25]. In these rural populations, CL is expressed as a single, rounded, and painless ulcer mainly located in the ear where it tends to become chronic if untreated [25]. At the time of writing, the annual incidence has been estimated at ~508/100,000 inhabitants [26]. However, given that 18.9–27.6% of the cases are asymptomatic, actual incidence in active foci could even be higher [22,25]. Regardless of the actual figures, CL control measures implemented in affected communities by the Mexican ministry of health, continue to be based on endorsing prevention via suggesting appropriate clothing for risk-prone activities and occupations [27].

The enzootic cycle of *Le. mexicana* is seasonal, beginning after the rainy period and persisting for approximately five months [28]. There, three wild rodent species act as primary reservoirs, *Ototylomys phyllotys* [29], *Heteromys gaumeri* [30] and *Peromyscus yucatanicus* [30–32]. Research on CL has so far identified at least eight sand fly species which could play an important role as vectors [23,33–36]. Among these sand flies, *Bichromomyia. olmeca olmeca* [37] is the only vector proven to be both infected by *Leishmania spp*. parasites and transmit them. However, this sand fly is considered more zoophilic than anthropophilic, as it is collected more frequently in traps baited with rodents than those baited with humans [33,34,38–40]. In addition, *Bi. olmeca* populations are smaller in disturbed landscapes [39,40]. In contrast, the sand fly *Lutzomyia. cruciata* (Coquillett), another suspected important vector, with confirmed high *Leishmania* infection prevalence [23,33,34], has higher preference for feeding on humans, [23,38], and appears to be more tolerant to human-dominated landscapes in the region, compared with *Bi. olmeca* [41–43].

Given the historical and contemporary epidemiological pattern of CL in the Yucatan Peninsula, exploring and understanding the mechanisms that regulate infection prevalence and incidence is still necessary. In this sense, mathematical models have been proposed previously to describe CL dynamics in the Americas, focusing on: 1) developing analytical expressions for the basic reproductive number [44]; 2) estimate critical epidemiological parameters from data at different rural localities in Venezuela [45]; and 3) analyze the dynamic properties of CL to test whether certain mitigation strategies could fail in Peru [46]. Mathematical models can provide insights to identify which prevention and control strategies are more effective and identify knowledge gaps to focus future research efforts [47]. Therefore, in this study, we present a simple mathematical model to describe the transmission dynamics of CL in the Yucatan Peninsula with the participation of two well-documented vectors (the sand fly species *Bi. olmeca* and *Lu. cruciata*) with different feeding behaviors and responses to the type of available bloodmeals from reservoir hosts (*O. phyllotis*, *P. yucatanicus* and *H. gaumeri*), and with humans as incidental hosts. Our study represents the first attempt to model the transmission dynamics of one of the most neglected but prevalent diseases in areas of high poverty and marginalization in the Yucatan Peninsula.

## Methods

### Model setting

We propose a CL transmission model comprising a rodent reservoir species, two sand fly vectors, *Lu. cruciata* and *Bi. olmeca*, and humans as incidental hosts with a set of nine ordinary differential equations (two equations for each vector species, two for reservoirs and three for humans). Among the rodent reservoir and the sand fly vectors, CL has Susceptible-Infected (*SI*) without recovery epidemiological dynamics. In turn, human CL follows Susceptible-Exposed-Infected-Susceptible (*SEIS*) dynamics. Transmission from rodents to sand flies, and from sand flies to rodents and humans is assumed to be frequency dependent. Furthermore, we assume that the two sand fly vectors are the most relevant species for the transmission of CL.

In humans, once CL infection occurs, there is an incubation period of 20 days before the onset of clinical symptoms. Afterwards, humans recover in approximately 21 days without protective immunity and re-enter the susceptible population. During the infectious period, we assume that humans are not capable of transmitting the infection neither to vectors nor to other humans, which is based on the most recent evidence of leishmaniasis epidemiology in the Yucatán Peninsula [25,48].

To represent the effect of available bloodmeals on sand fly species, we hypothesise that both human and rodent population sizes determine their carrying capacities in a logistic growth fashion. The second consequence of these different feeding ecologies is the variability of transmission among the two sand fly species.

**Model equations.** The population of each vector species $i$ ($V_i$, with $i = 1$ for *Lu. cruciata*, and $i = 2$ for *Bi. olmeca*) is divided into susceptible (*S*) and infected (*I*) compartments (Fig 1). The $S \rightarrow I$ transition depends on the probability that among its daily meals a vector bites an infected rodent reservoir. Hence the denominator of the frequency dependent transmission term (eqn. 1a) accounts for the proportion of meals taken from both humans and rodents, where subscripts $N$ indicate the entire population, including the different epidemiological compartments.

Regarding their population dynamics, the model assumes that both $S$ and $I$ vectors are equally fertile, and that the carrying capacity of vectors ($K_{V_i}$, eqn. 1c) increases linearly with rodent ($R_N$) and human ($H_N$) total population densities, with vector species-specific constants, $a_i$ and $b_i$, representing the number of sand flies that each rodent or human individual usually maintains alive, respectively [43]. Therefore, the equations describing these dynamics are:

$$\frac{dV_{i,S}}{dt} = r_V V_{i,N} \left( 1 - \frac{V_{i,N}}{K_{V_i}} \right) - \beta_V p_i \frac{V_{i,S} R_I}{p_i R_N + (1 - p_i) H_N} - \mu_V V_{i,S} \tag{1a}$$

$$\frac{dV_{i,I}}{dt} = \beta_V p_i \frac{V_{i,S} R_I}{p_i R_N + (1 - p_i) H_N} - \mu_V V_{i,I} \tag{1b}$$

$$K_{V_i} = a_i R_N + b_i H_N \tag{1c}$$

Tables 1–3 present the state variables and parameters associated with these equations, respectively.

Similarly to the vector population dynamics, the rodent dynamics are assumed to be unaffected by infection and all individuals are born susceptible. Epidemiologically, rodents are also divided into susceptible and infected, and the $S \rightarrow I$ transition after a bite from an infected vector is also frequency-dependent. The transmission rate from each vector species is the product of a general transmission rate ($\beta_R$) and the proportion of bites taken by vector species $i$ from rodents ($1 - p_i$). These dynamics are represented by the equations:

$$\frac{R_S}{dt} = r_R R_N \left( 1 - \frac{R_N}{K_R} \right) - \beta_R R_S \sum_{i=1}^{2} \frac{p_i V_{i,I}}{V_{i,N}} - \mu_R R_S \tag{2a}$$

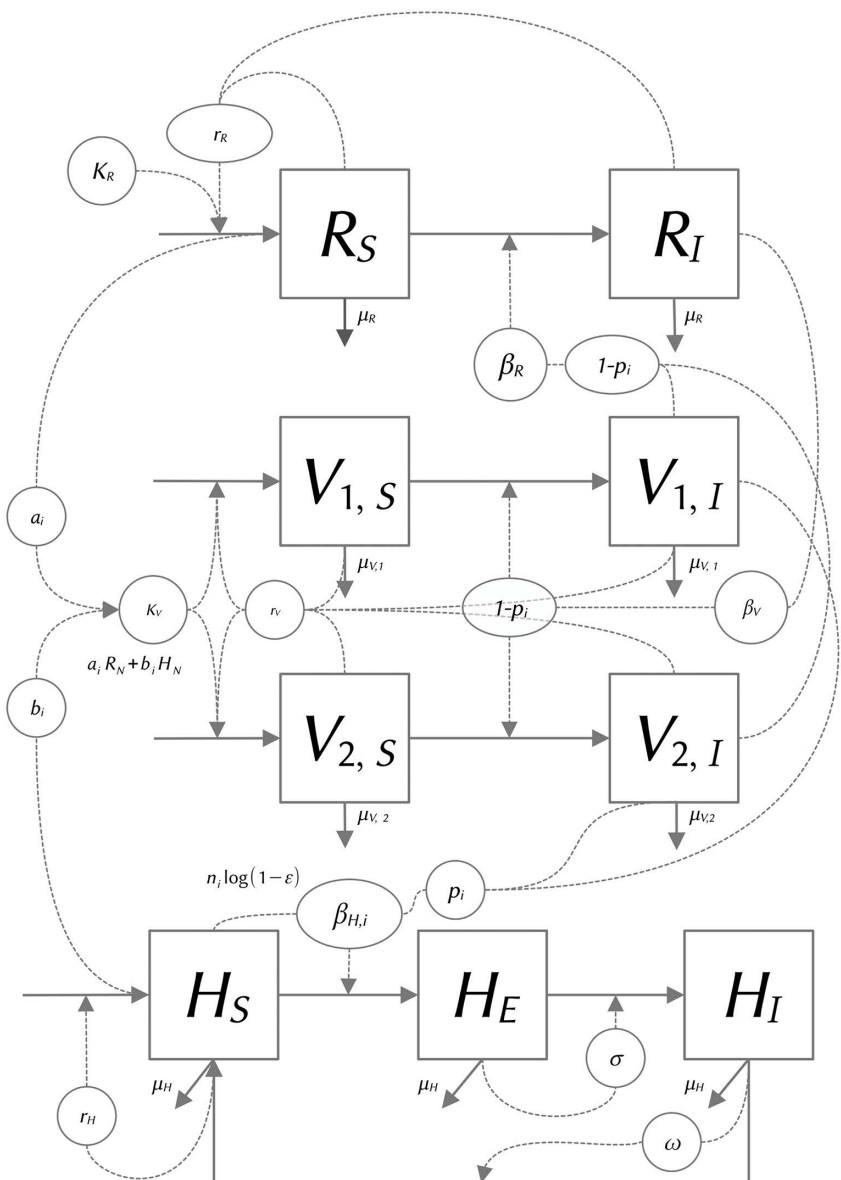

**Fig 1. Flow diagram of the model.** Square boxes correspond to state variables, while circles indicate model parameters. Solid arrows indicate in or outflows from each compartment, while dashed lines or arrows indicate the flows that are regulated by parameters interacting with the state variables connected to them. $R$ = reservoir host, $V_i$ = vector species, with $i$ = 1, *Lu. cruciata*, and $i$ = 2, *B. olmeca*, and $H$ = humans. Subindices $S$ = susceptible, $E$ = exposed. and $I$ = infected, identify the epidemiological status of each compartment.

$$\frac{R_I}{dt} = \beta_R R_S \sum_{i=1}^{2} \frac{p_i V_{i,I}}{V_{i,N}} - \mu_R R_I$$

(2b)

The population dynamics parameters $r_R$ and $K_R$ were estimated from field data (details below), and the latter was assumed to remain constant.

**Table 1. State variables associated to the cutaneous leishmaniasis transmission model, along with their initial values used in the simulations.** Epidemiological compartments: *S* = susceptible, *E* = exposed, and *I* = infected.

| State variable | Description | Epidemiological compartments | Initial value | | |
|---|---|---|---|---|---|
| | | | **S** | **E** | **I** |
| $V_1$ | Vector species *Lu. cruciata* | S, I | $K_1$ | – | 0 |
| $V_2$ | Vector species *Bi. olmeca* | S, I | $K_2$ | – | 0 |
| R | Reservoir hosts *H. gaumeri*, *O. phyllotis* and *P. yucatanicus* | S, I | 1499 | – | 1 |
| H | Humans | S, E, I | 60-600 | 0 | 0 |

**Table 2. Parameters associated to the cutaneous leishmaniasis transmission model, along with the fixed values used in the simulations and the source of these values.** i = 1 corresponds to vector species *Lu. cruciata* and i = 2 to vector species *Bi. olmeca*.

| Parameter | Description | Value | | Unit | Source |
|---|---|---|---|---|---|
| | | **i = 1** | **i = 2** | | |
| $r_V$ | Vector population growth rate | 0.22 | | day$^{-1}$ | [49] |
| $\mu_V$ | Natural vector mortality rate | 0.067 | | day$^{-1}$ | [49] |
| $a_i$ | Number of vector individuals per reservoir host | 0.14 | 0.5 | rodent$^{-1}$ | [43] |
| $b_i$ | Number of vector individuals per human | 14 | 2.5 | human$^{-1}$ | [43] |
| $\beta_V$ | Reservoir - vector transmission rate | 0.178 | | day$^{-1}$ | [50] |
| $p_i$ | Proportion of vector bites on rodents | 0.01 | 0.2 | dimensionless | Skin surface area[†] |
| $n_i$ | Average number of bites on humans | 4 | 0.83 | vector$^{-1}$ | [43] |
| $\varepsilon$ | Transmission efficiency | 0.034[††] | | Probability | [45] |
| $\beta_{H,i}$ | Vector - human transmission rate | $-n_i \log(1-\bar{\varepsilon})$[‡] | | day$^{-1}$ | [45] |
| $\beta_R$ | Vector - rodent transmission rate | $-n_i \log(1-\bar{\varepsilon})$[‡‡] | | day$^{-1}$ | [45] |
| $\sigma$ | Incubation period | 0.05 | | day$^{-1}$ | [51] |
| $\omega$ | Recovery rate for humans | 0.047 | | day$^{-1}$ | [51] |
| $r_R$ | Reservoir population growth rate | 0.03 | | day$^{-1}$ | [52–54] |
| $\mu_R$ | Natural reservoir mortality | 0.005 | | day$^{-1}$ | [55][‡‡‡] |
| $K_R$ | Reservoir carrying capacity | 30 | | Individuals Ha$^{-1}$ | [52–54] |

[†]Surface area is approximately 100 × larger for humans than rodents; therefore, *Lu. cruciata* is assumed to bite randomly according to available skin surface area. *B. olmeca*, is assumed to attract 20 × more rodents than humans.

[††]Calculated from the different values of $\varepsilon$ = {0.01, 0.01, 0.01, 0.01, 0.03, 0.04, 0.13} [45]. In certain simulations we used evenly spaced values covering the entire interval.

[‡]Keeling and Rohani [56], p. 18.

[‡‡]Here $\bar{\varepsilon}$ is the average of the different $\varepsilon$ values.

[‡‡‡]The reported value is 100 days in temperate Britain for house mice. We assumed double average lifespan for tropical mice due to shorter and more productive winter times.

For humans, population growth is not restricted by carrying capacity, but by the balance between $r_H$ and $\mu_H$. The $S \rightarrow I$ transition is frequency dependent, with a vector species-specific contact rate and proportion of meals taken from humans as the reciprocal of meals taken from rodents ($p_i$). The epidemiological parameters controlling the transitions $E \rightarrow I$ and $I \rightarrow S$ are $\sigma$ and $\omega$ respectively. The dynamics for humans are represented by the following equations:

$$\frac{dH_S}{dt} = r_H H_N - \left( \sum_{i=1}^{2} \frac{\beta_{H,i}(1-p_i) V_{i,I}}{V_{i,N}} + \mu_H \right) H_S + \omega H_I$$

(3a)

**Table 3. Parameters of the sand fly population growth model derived from Castillo et al. [49].**

| Parameter | Meaning | Value | Units |
|---|---|---|---|
| $n$ | Number of eggs per female | 41.1 | Individuals $\cdot$ t$^{-1}$ |
| $p_v$ | Fraction of eggs that hatch | 0.22 | dimensionless |
| $m_1$ | Hatching rate of eggs | 0.089 | t$^{-1}$ |
| $m_2$ | Maturation rate of stage-one larvae | 0.03 | t$^{-1}$ |
| $m_3$ | Maturation rate of stage-two larvae | 0.108 | t$^{-1}$ |
| $\mu_2$ | Mortality rate of stage-one larve | 0.017 | t$^{-1}$ |
| $\mu_3$ | Mortality rate of stage-two larve | 0.053 | t$^{-1}$ |
| $\mu_4$ | Mortality rate of adults | 0.017 | t$^{-1}$ |

$$\frac{dH_E}{dt} = \left( \sum_{i=1}^{2} \frac{\beta_{H,i} \left(1 - p_i\right) V_{i,I}}{V_{i,N}} \right) H_S - \left(\mu_H + \sigma\right) H_E \tag{3b}$$

$$\frac{dH_I}{dt} = \sigma H_E - \left(\mu_H + \omega\right) H_I \tag{3c}$$

## Demographic parameters

The parameters in the model associated with the dynamics of the state variables were the vector and reservoir population growth rates, $r_{V,i}$ and $r_R$, respectively, and the reservoir carrying capacity $K_R$. We approximated their values as described below. The model includes further parameters for which we obtained their values from the literature (Table 1).

**Vector population dynamics.** To approximate a value for $r_{V,1}$, we used laboratory-based life tables published by [49] for *Lu. cruciata*; we assumed that the corresponding parameter value for *Bi. olmeca* in the model is the same. The demographic dynamics was described through a continuous-time matrix population projection model, where vector populations were structured by development stage: 1) egg, 2) larvae, 3) pupae, and 4) reproductive adult. The value of $r_V$ was the real part of the dominant eigenvalue of the 4 × 4 matrix $L$:

$$L = \begin{bmatrix} -m_1 & 0 & 0 & np_V \\ m_1 & -(m_2 + \mu_2) & 0 & 0 \\ 0 & m_2 & -(m_3 + \mu_3) & 0 \\ 0 & 0 & m_3 & -\mu_4 \end{bmatrix}$$

where $n$ is the average number of eggs deposited, $p_V$ is the fraction that hatches, $m_j$ are the maturation rates (1/time spent in development stage $j$, in days), and $\mu_j$ is the state-specific mortality rate (Table 2).

**Reservoir population dynamics.** To approximate values for $r_R$ and $K_R$, we used published demographic data on the three most common rodent species of the Yucatan Peninsula, *P. yucatanicus*, *H. gaumeri* and *O. yucatanensis* [52–54] (F). In all cases, data consists of a time series of population density estimates in quadrats of 1 ha, in separate locations across the state of Yucatan (Hernandez-Betancourt et al. 2004, 2006, 2008). These time series were divided into two phases: an *r*-dominated and a *K*-dominated. *K*-dominated phases were those where population densities were larger than the 50% of the maximum population density recorded, and *r*-dominated phases were those where population densities were smaller than 50%. Then, we calculated population change between time points and used the exponential model to solve for *r*:

$$r_t = \log(N_{t+1}) - \log(N_t)$$

In this way, we obtained a series of values for $r$, which we then averaged to obtain a point estimate to use in the transmission model; the values of $K_R$ were also averaged. Both values of $r_R$ and $K_R$ represent the average for all rodent species taken together.

### Epidemiological parameters

The parameters controlling transition $S \rightarrow I$ are the transmission rates $\beta_{V,i}$ from reservoir to vector species $i$, $\beta_R$ from vectors to reservoir and $\beta_{H,i}$ for vectors to humans ([Fig 1]). To estimate a range of values for $\beta_{H,i}$ we used the approach developed by [57], where:

$$\beta = -n \log(1 - \varepsilon),$$

with $n$ the daily number of vector-human contacts, and $\varepsilon$ the transmission efficiency, both taken from the literature ([Table 3]). For $\beta_R$, to use as much data from field studies we used the same values derived for humans (Ravinovich and Feliciangeli, 2004) but weighted by the proportion of bites on rodents. Also, in light of the absence of data for $\beta_{V,i}$, we used the transmission rates for visceral leishmaniasis of [50].

For the fraction of *Lu. cruciata* bites on rodents, $p_i$, we assumed that it was proportional to the fraction of skin surface area available to draw a blood meal among humans and rodents. For *Bi. olmeca*, $p_i$ is 20 times the value for *Lu. cruciata* (see Table 3 for epidemiological parameters).

### Simulation setup and scenarios

Simulation scenarios were designed to show the range of possible outcomes during leishmaniasis one transmission season on five months duration (~150 days), in response to those parameters for which fewer information is available, namely the transmission rates $\beta$ parameters. To begin, we assumed that all $\beta$'s are a function of the transmission efficiency $\varepsilon$ estimated for other leishmaniasis endemic areas in the Americas [45].

The initial population values used in these simulations were the approximate human population found in Xpujil of ~6000 people [58] (Tatem et al. 2017), Campeche, the township where CL epidemiology is best understood in the study region, assuming 1, 5 and 10% of the population exposed to *Le. mexicana* vectors via various occupational risk factors. The initial values of vector populations, $V_i(0)$, were calculated at equilibrium level with the formula for $K_{V_i}$ (eqn. 1c) and were zero in scenarios where either or both sand fly species were absent.

Finally, to identify how the different parameters affect the system's behaviour we performed a a general sensitivity analysis using the Sobol-Martinez technique. To generate parameter uncertainty, we used a uniform distribution with minimum and maximum values corresponding to default values +/- 20%, to match the local sensitivity analysis. These analyses were performed using the ODEsensitivity R package [59].

## Results

The final model comprised nine differential equations with 11 parameters for the disease dynamics among vector species, four for the rodent reservoir dynamics and six for the transmission to humans ([Fig 1]). The transmission dynamics represented by the model were highly influenced by the vector species present. The scenario in which only *Lu. cruciata* was present, prevalence in one transmission season only reached a maximum of ~0.3% among humans (60 people), or 3 cases per 1000 population, ~100 days after one infected rodent was introduced in a 100% susceptible population, and after this point in time, prevalence decreased ([Fig 2]). However, in the scenario with only *Bi. olmeca* present, prevalence continued increasing and reached its peak at the end of the 150 day simulation with a ~1% among humans. In the

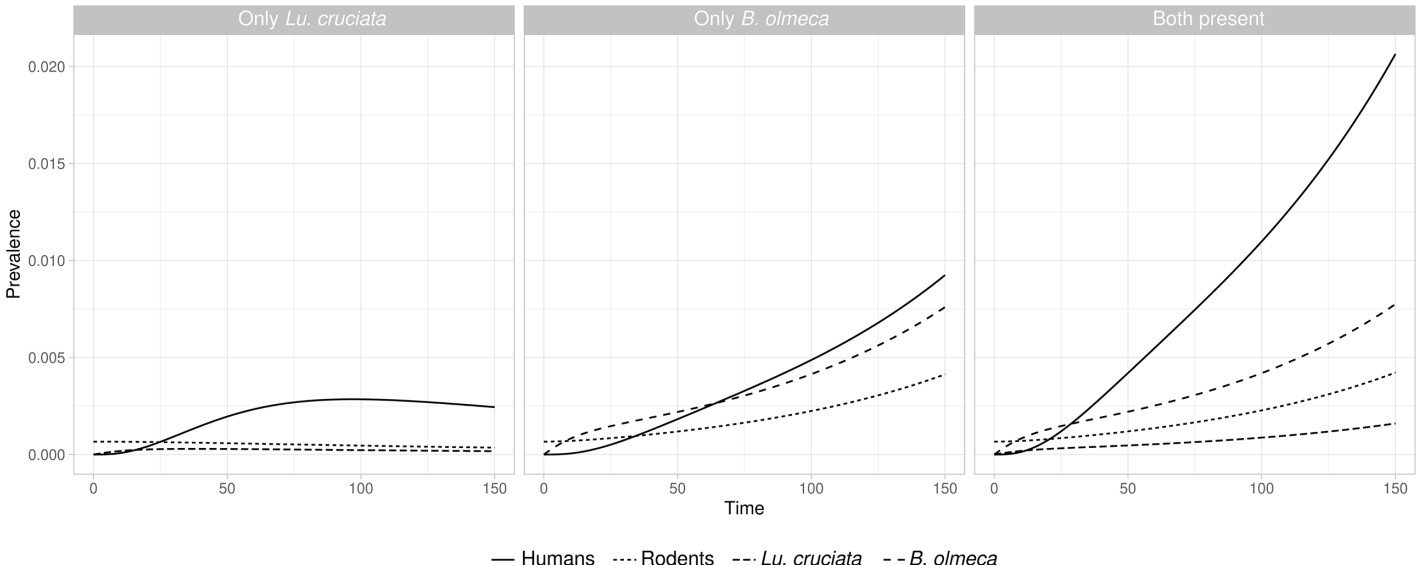

**Fig 2. Trajectory of infection prevalence in each species of the CL system in three scenarios based on the vector species present.** Initial conditions and variable parameters were $H(0) = 60$, $\varepsilon = 0.13$, $R = 1500$. $V_i(0)$ populations were calculated at equilibrium level with the formula for $K_i$, and were zero in scenarios where either are absent.

scenario with both vector species present, prevalence at the end of the simulation was ~2%. In the long term, prevalence stabilised around prevalence of 34% and 45% of the human population respectively (S1 Fig). These results indicate that with the parameter values used, *Bi. olmeca* is necessary for the maintenance of the parasite among reservoirs, while *Lu. cruciata* increases transmission of CL from rodents to humans (Fig 2). Likewise, the difference between the scenarios with and without *Lu. cruciata* on CL prevalence among rodents in the span of one transmission season was negligible.

CL prevalence among humans also depends on the size of the exposed human population, where understandably, higher prevalence is reached faster in smaller populations (Fig 3) due to: 1) immediate prevalence is higher once infection is introduced; and 2) human populations increase the carrying capacity of *Lu. cruciata*, increasing its population size more than that of *Bi. olmeca*. In the absence of human population growth, prevalence stabilises at around zero for both rodents and humans when only *Lu. cruciata* is present; it stabilises at around ~34, 46 and 52% among humans, rodents and *Bi. olmeca*, respectively, when only this sand fly is present; and around 44, 47, 53 and 18% among humans, rodents, *B. olmeca* and *Lu. cruciata*, respectively, when both species are present (S2 Fig).

Global sensitivity analysis revealed that leishmaniasis transmission to humans was most sensitive to the transmission rates involving rodents and *Bichromomyia olmeca*. Using the global sensitivity analysis, leishmaniasis transmission to humans was most sensitive to the transmission rates to and from rodents and *Bi. olmeca* ($\beta_R$ and $\beta_{V,2}$, respectively), the fraction of bites of *Bi. olmeca* on rodents ($p_2$), the transmission efficiency ($\varepsilon$), and the vector mortality rate ($\mu$). The contribution of humans to the carrying capacity of both sand fly species ($a_1$, $a_2$, $b_1$, and $b_2$) had a very weak effect on infection prevalence on humans (Fig 4; full Sobol sensitivity analyses are in S3 Fig).

## Discussion

Our simulations suggest that the presence of *Bi. olmeca* is necessary for *Le. mexicana* infection to persist among the rodent reservoir populations, when using the parameter values in Table 1. *Bi. olmeca* has long been considered one of the most important *Le. mexicana* vectors in the region (Biagi 1965), albeit with controversy. To address this issue, we included

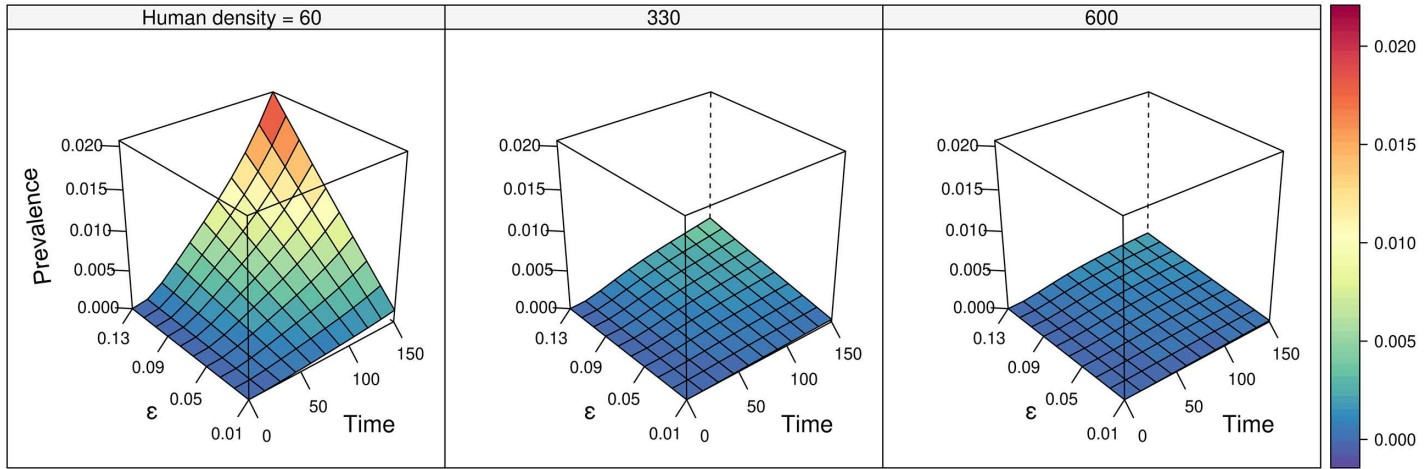

**Fig 3. Effect of increasing transmission efficiency (ε) on infection prevalence among humans, in each of the three possible scenarios of human population density.** Initial conditions and variable parameters were $H(0) = 60$, $\varepsilon = [0.01 - 0.13]$, $R_S(0) = 1500$, $R_I(0) = 1$. $V_i(0)$ populations were calculated at equilibrium level with the formula for $K_i = a_i R(0) + b_i H(0)$, and were zero in scenarios where either are absent.

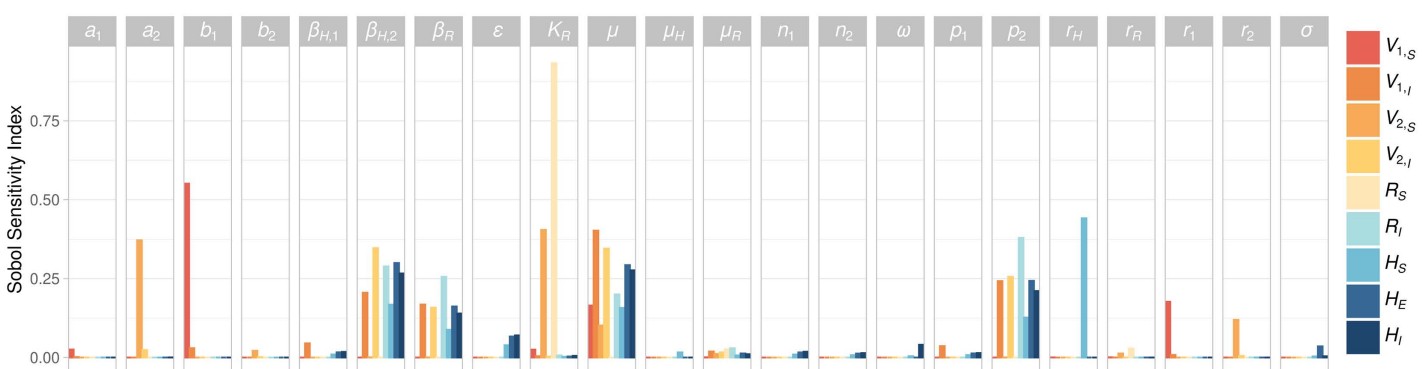

**Fig 4. Sobol sensitivity analysis of the system using a uniform distribution with minimum and maximum correspoding to the default values ±20%.** The height of the columns in the y-axis shows the Sobol total sensitivity index, which already takes into acccount the variance of all parameters, for each state variable (colour). The initial conditions were $H(0) = 60$, $R_S(0) = 1500$, $R_I(0) = 1$, $\varepsilon = 0.13$.

a second vector species, *Lu. cruciata*, found to be consistently present in leishmaniasis-endemic areas [33,34,39,41,60]. In contrast with *Bi. olmeca*, *Lu. cruciata* increases transmission to humans, as it does not have a preference for feeding on either humans or rodents. Hence, while *Bi. olmeca* maintains *Leishmania* infection, *Lu. cruciata*, via generalist feeding habits increases the range of infected hosts. This is evidenced by the difference observed between scenarios with, without, or including both vector species. Nevertheless, the absence of *Bi. olmeca* could be offset by an increase in the life expectancy of *Lu. cruciata*, as there would be more time for infected sand flies to contact susceptible rodents or humans. The end result is that the leishmaniasis transmission system in the Yucatan Peninsula is highly sensitive to the feeding ecology of the different sand fly species and the underlying carrying capacity determined by the availability and abundance of potential hosts as bloodmeals.

To the best of our knowledge, this is the first study to integrate the effect of sympatric *Leishmania* vector species on modelling its transmission dynamics. While previous studies have documented active infections on seven Phlebotominae (sand fly) species in the Yucatan Peninsula and Mexico [33,34,36,41,61], their role on infection maintenance and

transmission to humans had not been evaluated. On the contrary, most studies on transmission dynamics in the Americas have focused on the effect of reservoir species [44,60], but none of them in our study area. Compared with the cited studies, we acknowledge that our study area identifying system properties under the available evidence is highly determinant of the levels of transmission to humans. As this approach stems from the current level of understanding and data quality, a comprehensive research agenda we identify is resolving the role of vector and reservoir diversity on the leishmaniasis transmission system throughout its distribution in the Americas.

Even though the modelling exercise presented here has rendered novel insights, these should be taken with caution, the main reason being that knowledge on sand fly feeding preferences and demographic parameters remains limited, even with ~60 years of medical entomology research on leishmaniasis vectors in the study area [62]. Strinkingly, studies in the Yucatan Peninsula largely continue to focus on estimating infection prevalence, while profiles on feeding ecology are still poorly described for either *Bi. olmeca* or *Lu. cruciata.* Also, *Lu. cruciata* is the only sand fly species for which there is life history information in Mexico [49]. This undoubtedly limits the parameterization of our model and the interpretation of its results. A natural solution to this problem would be to generate basic feeding and population ecology information for *Bi. olmeca*, both of which will further help to understand the impact of this neglected tropical disease.

In a similar way that the vector feeding ecology affects leishmaniasis maintenance and transmission to humans, the diversity of reservoir species influences both epidemiological phenomena. In Venezuela, for instance, dogs and donkeys are its main reservoirs, but the latter increase leishmaniasis' reproductive number more than dogs [60]. Such studies evidence that introduced domestic species can become the most important reservoirs of human leishmaniasis. This is possible by a transmission cycle separate from the sylvatic (with native wild mammal reservoirs; [63]). In the Yucatan Peninsula, the full range of reservoir species have not been identified yet, although there are clinical leishmaniasis cases involving domestic dogs and cats [64]. This is important because isolated evidence has emerged of human infection within rural communities where *Bi. olmeca* populations are small and activities traditionally considered risk factors are not practiced (e.g., activities in the forest, [23,65]). Above, we highlighted the importance of life expectancy of both reservoir and vector species on maintaining infection. In this sense, and under our model system, human leishmaniasis in areas were *Bi. olmeca* populations are scarce may indicate the potential presence of alternative reservoir hosts with longer life expectancy such as dogs and cats, where *Lu. cruciata* may be able to maintain the transmission cycle. Non-sylvatic cycles imply that *Leishmania spp.* can infect multiple domestic animals in the Americas, further indicating that the diversity of the Phlebotominae feed sources are an important source of *Leishmania spp.*'s adaptation to new domestic hosts [66,67]. The process of adaptation to new hosts implicates that filling the knowledge gaps on feeding ecology, demography and reservoir host diversity is critical for disease management, as the preventive and mitigation strategies are radically different for zoonoses than diseases with human-to-human transmission.

Leishmaniasis in Europe, Asia and Africa is an anthropozoonosis, whereas in the Americas it is still considered a zoonosis where the main risk factor is carrying out forest-related activities (i.e., agriculture, catlle raising, etc), making it a significant occupational disease. Currently, there is no evidence that humans are *Leishmania spp.* reservoirs in the Neotropics, although the high prevalence of asymptomatic human cases in the Yucatan Peninsula [22,26], and the frequent detection of live parasites in these [68] begs the question whether humans are in fact able to transmit *Le. mexicana* to vectors. If humans are proven to be competent reservoirs, public health policies should then target human-to-human and zoonotic transmission. In terms of our model, this novel characteristic would change the role of *Lu. cruciata* from driving zoonotic transmission and potential dilution (e.g., [69]) to maintaining *Le. mexicana* populations among humans and domestic reservoir hosts.

Leishmaniasis incidence among humans is still poorly characterised. Early analyses have found 508 cases per 100,000 inhabitants, but prevalence as high as 43% [26]. The prevalence and incidence figures produced by our model are nowhere near those values. First because we did not seek to estimate them but to show the effect of feeding preferences of different vector species while highlighting its epidemiological implications. As shown in the supplementary materials, if

the simulation is let run for many more time units, the prevalence among humans is ~45% (S2 Fig), although this does not account for seasonality for the above reasons. Policy-wise, we do not recommend using our model for decision making, but to incentivise reports and the funding of epidemiological studies to clarify if such figures are still patent among populations at risk.

As seen above, the role of *Lu. cruciata* and other sand fly species should not be minimised in spite of current evidence, suggesting that *Bi. olmeca* maintains *Le. mexicana* infection (with the parameters in Table 1). The high prevalence observed in *Lu. cruciata*, its widespread distribution and high adaptability to human-modified landscapes [39,42] where domestic animal bloodmeals are abundant, are compelling arguments to consider it the pivotal vector towards a non-sylvatic transmission cycle. Further empirical evidence of the possibility of non-sylvatic cycles has been recorded in recent outbreaks associated with sand fly species not considered in this study in Yucatán [39,42]. The ability of *Le. mexicana* to infect a wide range of vectors and both reservoir and dead-end hosts has evolutionary consequences. For instance, we infer that on the one hand, specialist vectors like *Bi. olmeca* act as efficient maintenance vectors among reservoirs; on the other, generalists like *Lu. cruciata* help *Le. mexicana* to increase its range of hosts. These properties acquire even more relevance under the current process of intense land conversion across the study area [70], which increases exposure of humans and domestic animals to *Le. mexicana*.

As is evident up to this point, there are several uncertainties in how *Leishmania spp*. parasites are maintained in the study area where we identify a large set of open questions relevant from a public health, ecological and environmental point of view: 1) lack of knowledge about specific competition between vector species; 2) inability to describe the response of the different sand fly species' feeding preference to the abundance and availability of different bloodmeals; 3) the response of sand flies and rodent reservoirs to land use change and human population growth; 4) the relationship between sand fly species and domestic or synanthropic wild mammals; 5) the role of humans in the transmission dynamics, mainly the asymptomatic population with viable *Leishmania spp*. parasites; and 6) the effect that all of the previous limitations have on contact and transmission rates. Previous analyses show that leishmaniasis' epidemiological parameters can be very variable, even among neighboring communities [45], which may be caused by some of the phenomena listed above and thus speak of the necessity to comprehensively characterize the eco-epidemiology of leishmaniasis

## Conclusions

Our results highlight the importance of the feeding ecology of sand fly species on the *Le. mexicana* transmission dynamics in the Yucatán Peninsula. While *Bi. olmeca* is essential for maintaining the infection among reservoirs, *Lu. cruciata* and other generalist species drive zoonotic transmission and dilution among non-competent dead-end hosts, which could in turn expand the parasite's range of reservoir hosts. By adapting to these new domestic hosts, the generalist species could act as ecological bridges, facilitating the emergence of novel transmission cycles and increasing the risk of infection in humans. The land conversion process currently accelerating across the study area increases the risk of emergence of non-sylvatic transmission cycles. Therefore, increasing our knowledge basis to understand the process of *Le. mexicana* transmission dynamics, to generate applicable solutions for the public health challenges the Yucatan Peninsula is facing is of utmost importance.

## Supporting information

**S1 Fig. Timelines of the three rodent species population densities used to estimate parameters $r_R$, $K_R$ and $m_R$.**
(EPS)

**S2 Fig. Model simulation for 2500 days to obtain prevalence at stability conditions.**
(EPS)

**S3 Fig. Sobol indices for the full 150 day simulation period.** Top panel shows First order indices obtained by keeping all parameters fixed except the target parameter. The bottom panel shows total sensitivity, obtained by simultaneous Monte Carlo simulation of all parameters.
(EPS)

## Author contributions

**Conceptualization:** Gerardo Martin, Edgar J. González, Ana Celia Montes de Oca-Aguilar.

**Data curation:** Gerardo Martin, Edgar J. González, Ana Celia Montes de Oca-Aguilar.

**Formal analysis:** Gerardo Martin, Edgar J. González.

**Investigation:** Gerardo Martin, Edgar J. González, Ana Celia Montes de Oca-Aguilar.

**Methodology:** Gerardo Martin.

**Visualization:** Gerardo Martin.

**Writing – original draft:** Gerardo Martin, Edgar J. González, Ana Celia Montes de Oca-Aguilar.

**Writing – review & editing:** Elsy Nallelli Loría-Cervera.

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
