## [Decision Letter · Decision Letter 0]

12 Aug 2025

Endemic and zoonotic cycles of cutaneous leishmaniasis depend on vector feeding preferences: an epidemiological model for Southeastern Mexico

Dear Dr. Montes de Oca-Aguilar,

Thank you for submitting your manuscript to PLOS Neglected Tropical Diseases. After careful consideration, we feel that it has merit but does not fully meet PLOS Neglected Tropical Diseases's publication criteria as it currently stands. Therefore, we invite you to submit a revised version of the manuscript that addresses the points raised during the review process.

Please submit your revised manuscript within 60 days Oct 11 2025 11:59PM. If you will need more time than this to complete your revisions, please reply to this message or contact the journal office at plosntds@plos.org. Please include the following items when submitting your revised manuscript:

We look forward to receiving your revised manuscript.

Kind regards,

Indrajit Ghosh

Academic Editor

Audrey Lenhart

Section Editor

Shaden Kamhawi

co-Editor-in-Chief

Paul Brindley

co-Editor-in-Chief

**Journal Requirements:**

At this stage, the following Authors/Authors require contributions: Gerardo Martín, Edgar J. González, Nalleli Loría-Cervera, and Ana Celia Montes de Oca-Aguilar. Please ensure that the full contributions of each author are acknowledged in the "Add/Edit/Remove Authors" section of our submission form.

3) When completing the data availability statement of the submission form, you indicated that you will make your data available on acceptance. We strongly recommend all authors decide on a data sharing plan before acceptance, as the process can be lengthy and hold up publication timelines. Please note that, though access restrictions are acceptable now, your entire data will need to be made freely accessible if your manuscript is accepted for publication. This policy applies to all data except where public deposition would breach compliance with the protocol approved by your research ethics board. If you are unable to adhere to our open data policy, please kindly revise your statement to explain your reasoning and we will seek the editor's input on an exemption. Please be assured that, once you have provided your new statement, the assessment of your exemption will not hold up the peer review process.

**Reviewers' Comments:**

Reviewer's Responses to Questions

**Key Review Criteria Required for Acceptance?**

**Methods:**

-Are the objectives of the study clearly articulated with a clear testable hypothesis stated?

-Is the study design appropriate to address the stated objectives?

-Is the population clearly described and appropriate for the hypothesis being tested?

-Is the sample size sufficient to ensure adequate power to address the hypothesis being tested?

-Were correct statistical analysis used to support conclusions?

-Are there concerns about ethical or regulatory requirements being met?

Reviewer #1: Methods should be more detailed and aligned with the objectives of this work. The model must be adjusted according to the suggestions in the reviewer attachment. Additionally, the parameters used for this simulation should have appropriate references included.

Reviewer #2: This manuscript develops a compartmental model for cutaneous leishmaniasis (CL) in the Yucatán Peninsula with two vector species (Lutzomyia cruciata and Bichromomyia olmeca), a rodent reservoir, and humans as incidental hosts. Vectors and rodents follow SI dynamics; humans follow SEIS dynamics. The authors explore scenarios and conclude that Bi. Olmeca maintains infection in reservoirs while Lu. Cruciata drives zoonotic spillover to humans. Going through the manuscript, I have several comments and suggestions.

1. In Table 2, the vector-to-human and vector-to-rodent transmission rates are defined as β_{H,i} = n_i * ln(1 − ε) and β_R = n_i * ln(1 − ε). As written, ln(1 − ε) ≤ 0, making β negative. Please verify and correct throughout (including any code/simulations), and update results if necessary.

2. β_R is built from the average number of bites on humans (n_i) rather than bites on rodents (which should depend on host availability and p_i). Please explain the motivation behind this assumption in more detail.

3. The model considered rodents as a reservoir; however, dogs can also act as reservoir (https://doi.org/10.3390/tropicalmed3020043). Please give supporting statements to discard all other reservoirs other than rodents.

4. The model assumes infectious humans cannot infect susceptible vectors. Please (a) justify the zero-infectivity assumption with citations specific to Le. mexicana in the study area, or (b) add a sensitivity/scenario where humans have low but non-zero infectivity to vectors.

5. The Introduction and prior work emphasize seasonal transmission in the region, but the model lacks seasonal forcing (https://journals.plos.org/plosntds/article?id=10.1371/journal.pntd.0003283). Please justify this assumption.

6. Simulations are qualitative. For policy relevance, please compare predicted prevalence or incidence trajectories to data from Xpujil or nearby foci (even if approximate).

7. Other minor comments:

a. “sensible to transmission rates” → “sensitive to transmission rates” (Abstract/Results).

b. “hypothesis that” → “hypothesize that” (Methods).

c. Species names: ensure consistent, correct spelling and italics: Peromyscus yucatanicus (not “Peromiscus”), Ototylomys phyllotis, Heteromys gaumeri. I noticed inconsistencies in Table 1 species list.

d. Use either Bichromomyia olmeca or B. olmeca olmeca consistently.

e. Units: make p_i unit-free; confirm units for n_i and βs

f. Line 230, the equation is not readable.

**Results:**

-Does the analysis presented match the analysis plan?

-Are the results clearly and completely presented?

-Are the figures (Tables, Images) of sufficient quality for clarity?

Reviewer #1: The figure is not clearly visible; it should be of higher resolution.

Reviewer #2: (No Response)

**Conclusions:**

-Are the conclusions supported by the data presented?

-Are the limitations of analysis clearly described?

-Do the authors discuss how these data can be helpful to advance our understanding of the topic under study?

-Is public health relevance addressed?

Reviewer #1: Conclusion is ok.

Reviewer #2: (No Response)

**Editorial and Data Presentation Modifications?**

Reviewer #1: Major revision needed before publication.

Reviewer #2: (No Response)

**Summary and General Comments:**

Reviewer #1: Please see the reviewer attachment

Reviewer #2: (No Response)

PLOS authors have the option to publish the peer review history of their article (what does this mean? ). If published, this will include your full peer review and any attached files.

**Do you want your identity to be public for this peer review?** For information about this choice, including consent withdrawal, please see our Privacy Policy .

Reviewer #1: No

Reviewer #2: No

**Figure resubmission:**

**Reproducibility:**



---

## [Decision Letter · Decision Letter 1]

20 Nov 2025

Dear Dr. Martin,

We are pleased to inform you that your manuscript 'Endemic and zoonotic cycles of cutaneous leishmaniasis depend on vector feeding preferences: an epidemiological model for Southeastern Mexico' has been provisionally accepted for publication in PLOS Neglected Tropical Diseases.

Best regards,

Indrajit Ghosh

Academic Editor

Audrey Lenhart

Section Editor

Shaden Kamhawi

co-Editor-in-Chief

Paul Brindley

co-Editor-in-Chief

Reviewer's Responses to Questions

**Key Review Criteria Required for Acceptance?**

**Methods**

-Are the objectives of the study clearly articulated with a clear testable hypothesis stated?

-Is the study design appropriate to address the stated objectives?

-Is the population clearly described and appropriate for the hypothesis being tested?

-Is the sample size sufficient to ensure adequate power to address the hypothesis being tested?

-Were correct statistical analysis used to support conclusions?

-Are there concerns about ethical or regulatory requirements being met?

Reviewer #1: All the above points are addressed properly

Reviewer #3: (No Response)

**Results**

-Does the analysis presented match the analysis plan?

-Are the results clearly and completely presented?

-Are the figures (Tables, Images) of sufficient quality for clarity?

Reviewer #1: All the above points are addressed properly

Reviewer #3: (No Response)

**Conclusions**

-Are the conclusions supported by the data presented?

-Are the limitations of analysis clearly described?

-Do the authors discuss how these data can be helpful to advance our understanding of the topic under study?

-Is public health relevance addressed?

Reviewer #1: All the above points are addressed properly

Reviewer #3: (No Response)

**Editorial and Data Presentation Modifications?**

Reviewer #1: Accept

Reviewer #3: (No Response)

**Summary and General Comments**

Reviewer #1: All my previous queries are addressed properly

Reviewer #3: (No Response)

PLOS authors have the option to publish the peer review history of their article (what does this mean? ). If published, this will include your full peer review and any attached files.

**Do you want your identity to be public for this peer review?** For information about this choice, including consent withdrawal, please see our Privacy Policy .

Reviewer #1: No

Reviewer #3: No

---

## [Editor Report · Acceptance letter]

Dear Dr. Martin,

We are delighted to inform you that your manuscript, "Endemic and zoonotic cycles of cutaneous leishmaniasis depend on vector feeding preferences: an epidemiological model for Southeastern Mexico," has been formally accepted for publication in PLOS Neglected Tropical Diseases.

Best regards,

Shaden Kamhawi

co-Editor-in-Chief

Paul Brindley

co-Editor-in-Chief
